# Ovarian Follicle Depletion Induced by Chemotherapy and the Investigational Stages of Potential Fertility-Protective Treatments—A Review

**DOI:** 10.3390/ijms20194720

**Published:** 2019-09-23

**Authors:** Xia Hao, Amandine Anastácio, Kui Liu, Kenny A. Rodriguez-Wallberg

**Affiliations:** 1Department of Oncology-Pathology, Karolinska Institutet, SE-171 76 Stockholm, Sweden; xia.hao@ki.se (X.H.); amandine.anastacio@ki.se (A.A.); 2Laboratory of Translational Fertility Preservation, BioClinicum, SE-171 64 Stockholm, Sweden; 3Department of Obstetrics and Gynecology, The University of Hong Kong, Hong Kong, China; kliugc@hku.hk; 4Director Center for Meiosis and Germ Cell Development, Shandong University, Jinan 250001, China; 5Department of Reproductive Medicine, Division of Gynecology and Reproduction, Karolinska University Hospital, SE-141 86 Stockholm, Sweden

**Keywords:** cancer, chemotherapy, ovarian follicle depletion, infertility, ovarian protection

## Abstract

Ovarian follicle pool depletion, infertility, and premature menopause are all known sequelae of cancer treatment that negatively impact the quality of life of young cancer survivors. The mechanisms involved in this undesired iatrogenic ovarian damage have been intensively studied, but many of them remain unclear. Several chemotherapeutic drugs have been shown to induce direct and indirect DNA-damage and/or cellular stress, which are often followed by apoptosis and/or autophagy. Damage to the ovarian micro-vessel network induced by chemotherapeutic agents also seems to contribute to ovarian dysfunction. Another proposed mechanism behind ovarian follicle pool depletion is the overactivation of primordial follicles from the quiescent pool; however, current experimental data are inconsistent regarding these effects. There is great interest in characterizing the mechanisms involved in ovarian damage because this might lead to the identification of potentially protective substances as possible future therapeutics. Research in this field is still at an experimental stage, and further investigations are needed to develop effective and individualized treatments for clinical application. This review provides an overview of the current knowledge and the proposed hypothesis behind chemotherapy-induced ovarian damage, as well as current knowledge on possible co-treatments that might protect the ovary and the follicles from such damages.

## 1. Introduction

Effective screening and early diagnosis combined with improved cancer therapies have contributed to increased long-term survival after cancer treatment. However, chemotherapy and radiotherapy in both sexes have numerous side effects that dramatically decrease quality of life among cancer survivors. One of the most difficult long-term side effects that cancer survivors have to cope with is infertility, which in females is mainly due to the ovarian damage and subsequent premature ovarian failure (POF), a common sequel of cancer treatment.

The risk of subsequent infertility in young cancer patients is associated with the treatment modalities that would be applied. Regarding chemotherapeutic agents, especially those with alkylating properties have demonstrated a high gonadal toxicity in females, causing follicular depletion, amenorrhea, and infertility. Alkylating drugs are included in most cancer treatment protocols and they are also used to treat benign autoimmune diseases. In high doses, alkylating drugs are also used for conditioning prior to hematologic stem cell transplantation for treating either malignant or benign conditions [1,2,3].

In response to the needs of a young cancer population undergoing highly gonadotoxic treatments, medical strategies to overcome infertility have been developed, and those are offered in clinical practice. [3]. These are summarized in Figure 1.

For female patients, methods such as the cryopreservation of oocytes, embryos, or ovarian tissue may be offered [1,3,4,5,6]. The development of techniques for fertility preservation has rapidly evolved into a fruitful research field with a focus not only on patients of reproductive age, but also on children and adolescents [7], and programs for fertility preservation have been established in many countries and have reported a continuous increase in numbers of patients undergoing such procedures [8,9,10,11,12]. However, female fertility preservation options are challenging in several clinical situations, such as the case of pre-pubertal girls in whom mature oocyte cryopreservation is not feasible or when time constraints do not allow for hormonal stimulation aiming at mature egg retrieval in adult women. Ovarian tissue cryopreservation might be a feasible option because it does not require such time, but it is problematic when the patients are diagnosed with ovarian carcinomas or with malignancies that might spread to the ovaries, thus precluding future auto-transplantation of the ovarian tissue due to the risk of re-introducing malignant cells [4,5,13,14,15]. Additionally, the methods for fertility preservation are expensive and unaffordable for many patients worldwide [16,17], and they are also invasive. However, even in countries with established tax-funded healthcare available to the entire population, women seem to be at a disadvantage compared to men with regards to information about methods of fertility preservation [18].

Furthermore, the ovary is not only responsible for fertility, but also an important organ that maintains the endocrinological balance of a woman until the onset of menopause, which naturally occurs in women around 50 years of age, but which can be accelerated or delayed by several comorbidities [19]. Thus, the investigations of ovarian protective treatments aiming at fertility preservation during cancer treatment are greatly needed. In order to develop such protective treatments, the mechanisms behind how chemotherapy damages the ovary have to be better understood. Many of the mechanisms involved remain unclear [20,21,22,23,24,25,26,27,28,29], but the two main theories that dominate this research field are that POF is the result of the induction of DNA damage and/or oxidative stress in the follicles that triggers apoptosis [21,22] or is the result of overactivation of the dormant primordial follicles [25,26]. Most of the research on these damaging mechanisms has been performed in experimental animal models, and the translation to clinical treatments might take years. This review summarizes current knowledge on the mechanisms involved in chemotherapy-induced ovarian damage as well as the investigational protective treatments aiming at fertility preservation.

## 2. Gametogenesis and Folliculogenesis

Anatomically, the ovary is divided into three regions, including an outer cortical region containing the germinal epithelium and follicles, a medullary region composed of connective tissue, myoid-like contractile cells, and interincisal cells and the hilum, that contains the blood vessels, lymphatic vessels, and nerves entering into the ovary. The entire ovary is surrounded by the albuginea capsule. The ovary performs two main functions that are correlated, namely, gametogenesis for reproduction and the production of steroid hormones and various factors that meet the body’s endocrinological needs [30].

Gametogenesis in humans starts during embryonic development in which a subpopulation of diploid and pluripotent cells called primordial germ cells (PGCs) migrate to the gonadal ridge. In females, the PGCs give rise to oogonia which were arrested at the diplotene stage of meiosis prophase I when surrounded by a layer of flattened somatic cells (pregranulosa cells) to form primordial follicles [31,32], and these primordial follicles form the ovarian reserve, also known as the follicle pool. Although the follicle pool reaches a peak of 6–7 million primordial follicles at 20 weeks of gestation, this number falls to 1–2 million at birth, and by the time the woman enters puberty only around 400,000 primordial follicles are remaining [33].

Once a girl enters puberty, waves of primordial follicles begin to be recruited into activation, which is irreversible. In activated follicles, the oocytes grow robustly, the surrounding pregranulosa cells differentiate into cuboidal granulosa cells and proliferate, developing through primary follicles and thereafter secondary follicles before acquiring an antral cavity. These follicles are in preantral stage, their development is mainly controlled by locally produced growth factors, and they may progress to the antral stage in absence of gonadotropins, as shown in patients with defective FSH-receptors [34,35]. When these follicles enter the antral stage, only a few of them can develop till the preovulatory stage, while most of them undergo atresia due to negative regulation from the dominant Graafian follicle. The growth of antral follicles is gonadotropin-dependent, regulated by feed-back responses between gonadotropin-releasing hormone (GnRH), follicle-stimulating hormone (FSH), luteinizing hormone (LH), and various growth factors [36,37]. Under the preovulatory LH gonadotropin surge, the dominant Graafian follicle ovulates to release a mature oocyte and the remaining theca and granulosa cells transform to a corpus luteum [34,38]. The different stages of folliculogenesis are illustrated in Figure 2.

This pattern of follicle recruitment and selection leads to a continuous and progressive decrease in the number of available primordial follicles in the follicle pool and a limited reproductive life span. Menopause occurs when the ovarian follicle pool is exhausted [39], thus the size and persistence of the ovarian reserve determine the length of reproductive life [37]. In addition to the natural causes of ovarian pool depletion, some external factors such as lifestyle factors, including using tobacco (even exposure to second-hand smoke) can accelerate this process, but this process is most dramatically and rapidly accelerated by exposure to radiation and different toxic agents [40,41,42,43,44].

### 2.1. Ovarian Reserve Dormancy

During recruitment, only some primordial follicles are activated and the rest are kept dormant in the resting pool. The dormancy of primordial follicles is maintained by the balance of inhibitory and stimulatory molecules [45,46,47]. Several molecules such as the forkhead box (FOX) proteins FOXL2 and FOXO3a, anti-Müllerian hormone (AMH), phosphatase and tensin homolog (PTEN), p27Kip1 (also known as cyclin-dependent kinase inhibitor 1B, kinase inhibitory protein 1, and p27), and tuberous sclerosis complex (TSC) have been identified as inhibitory factors (Figure 3).

FOXL2 is a forkhead domain/winged-helix transcription factor that has been shown to be an important transcription factor for ovarian development, folliculogenesis, follicle maintenance, and sex determination/maintenance in mammals [48,49]. In humans, carriers of *Foxl2* mutations present with a higher risk of POF [50]. In mice, FOXL2 is expressed in pregranulosa cells of primordial follicles, with a decreased expression in granulosa cells of preantral follicles [46,51]. Its role in pregranulosa cell differentiation and ovary maintenance was demonstrated using *Foxl2^lacZ^* homozygous mutant mice. In those mice, most of the primordial follicles were activated which was shown by the growth of oocytes, but the granulosa cells were blocked at the squamous to cuboidal transition and no secondary follicles were formed [52]. In the absence of functional granulosa cells, the activated follicles subsequently underwent apoptosis, resulting in oocyte atresia and progressive ovarian reserve depletion [52]. These findings indicate that FOXL2 in pregranulosa cells is essential for the maintainance of primordial follicle quiescence and suppression of initiation of oocyte growth [46].

FOXO3a is another transcription factor in the FOX protein family and is one of the downstream effectors of the PTEN/PI3K/Akt signaling pathway [53]. In *Foxo3a*-null mice, the primordial follicles were globally activated, and thus greater numbers of primary and secondary follicles were observed before the onset of sexual maturity compared to wild-type controls. Additionally, no normal primordial follicles were observed in postnatal day (PD)14 ovaries, and instead the primordial follicles contained large, misshapen oocytes with fragmented nuclei [54]. Moreover, transgenic mice overexpressing constitutively active FOXO3a showed a larger ovarian reserve compared to their wild-type littermates, demonstrating the role of FOXO3a as a guardian of the ovarian pool [55].

AMH is a member of the transforming growth factor-β superfamily and is produced by the granulosa cells of growing follicles from the secondary to the early antral stage [56,57]. It is considered to be a suppressive factor because it inhibits the transition from primordial to primary follicles [56,57,58]. In *Amh^−/−^* female mice, more preantral follicles and small antral follicles – and thus fewer primordial follicles – were observed compared to wild-type mice at the same age [59]. The addition of AMH to the culture medium has been shown to maintain the percentages of primordial follicles in PD4 rat ovaries after 10 days of culture, and similar results have been obtained in human ovarian cortical tissue culture [60].

PTEN was originally identified as a tumor suppressor due to its function as a negative regulator of PI3K signaling [61]. In the oocyte, PTEN is considered to play a role in maintaining the dormancy of primordial follicles [62], and premature activation of the entire ovarian pool resulting in POF in early adulthood is seen in Oo*Pten^−/−^* mice [63]. In addition, in vitro activation of primordial follicles has been observed when human or mouse ovarian tissue was treated with PTEN inhibitor, thus providing a novel way of activating primordial follicles in clinics [64,65,66].

p27Kip1 is expressed in the oocytes and granulosa cells of primordial, primary, and secondary follicles. In larger follicles, its expression is only detectable in granulosa cells. The ovaries of *p27^−/−^* mice had more developed follicles and fewer primordial follicles, resulting in follicle depletion in early adulthood compared to wild-type mice [67].

TSC1 and TSC2 are products of the tumor suppressor genes *TSC1* and *TSC2* [68]. TSC1 and TSC2 negatively regulate the activity of the mammalian target of rapamycin complex 1 (mTORC1), and deletion of *Tsc1* or *Tsc2* from mouse oocytes in primordial and further developed follicles has been shown to cause overactivation of the entire ovarian pool and subsequent POF in early adulthood [69,70].

### 2.2. Ovarian Reserve Maintenance

Under the influence of both intrinsic and extrinsic inhibitory factors, the majority of primordial follicles are maintained in a quiescent state. Even though they are dormant, various factors are required to keep them alive. The PTEN/PI3K-PDK1-Akt-S6K1-rpS6 and TSC/mTORC1-S6K1-rpS6 signaling pathways have both been shown to be essential for primordial follicle survival in the resting pool, especially their common downstream S6K1-rpS6 signaling factors [46,71].

To ensure primordial follicle survival, extracellular signals maintain a basal degree of PI3K activation in the oocyte [46]. The signaling mediated by PI3K converges first at PDK1 [72], which phosphorylates and thus activates Akt and S6K1. The kinase S6K1 then activates rpS6 [46,71]. S6K1-rpS6 signaling is involved in protein translation and ribosome biogenesis in oocytes [69,70,73], and in Oo*Rps6^−/−^* mice the ovaries were much smaller compared to controls at PD23, and at 8 weeks of age no follicular structures were observed [71].

In the TSC-mTORC1-S6K1-rpS6 signaling pathway, TSC activates mTORC1, which in turn activates S6K1 and rpS6. It has been hypothesized that deletion of *mTOR* in oocytes leads to accelerated primordial follicle death, and several experimental studies support this hypothesis [46,69,71].

The absence of *Pdk1* in oocytes has been shown to be directly related to POF with reduced numbers of healthy primordial follicles in PD35 Oo*Pdk1^−/−^* mice and the absence of primordial follicles at 8 weeks of age despite normal follicle survival in younger mice [71].

## 3. Chemotherapy-Induced Ovarian Damage

Chemotherapeutics are cytotoxic drugs used in cancer treatment due to their ability to inhibit tumor cell division and to trigger tumor cell death. Some of these drugs are also used for the treatment of chronic inflammatory diseases and other non-cancerous conditions. According to the targeted phases of the cell cycle, cytostatics are classified as cell cycle-nonspecific agents or cell cycle-specific agents. Cell cycle-nonspecific agents target cells in all phases of the cell cycle, including the G_0_ phase. They generally show a rapid effect with a linear dose-response curve [74]. Cell cycle-specific agents are only effective against cells in specific phases and do not target cells in G_0_ phase, and their effects tend to be schedule/time dependent and to not increase with dose after reaching a maximum level [74].

Different types of chemotherapeutic drugs show different levels of ovarian toxicity, and they target follicles at different developmental stages. The magnitude of ovarian toxicity is highest in alkylating agents, and it decreases in the order of platinum analogs, taxanes, plant alkaloids, anthracyclines, and anti-metabolites, which are generally considered to be non-gonadotoxic [42,75]. Because primordial follicles are quiescent, they are more sensitive to cell cycle-nonspecific agents such as alkylating agents and topoisomerase inhibitors, while anti-metabolites mainly affect the developing follicles without altering the dormant follicle pool [75,76]. In developing follicles, since their granulosa cells are proliferating, all cytostatics can damage them resulting in clinical symptoms such as temporary amenorrhea [75]. The types of cytostatics, representative drugs, and their anti-tumor mechanisms are summarized in Table 1.

The mechanisms behind primordial follicle loss induced by chemotherapy have been extensively investigated, but they remain largely unclear [20,21,22,23,24,25,26,27,28,29]. The reported mechanisms include a primary or a secondary effect on the primordial follicles resulting in depletion of the ovarian reserve. As their primary effect (Figure 4), chemotherapeutic agents can directly induce DNA damage (e.g. double-strand breaks (DSBs), inter- and intra-strand crosslinks, intercalation, and monoalkylation), which results in the activation of apoptosis and/or autophagy-related pathways [20,22,23,24,27], or they can indirectly cause DNA damage by increasing oxidative stress or by damaging the ovarian micro-vessel network, leading to cellular stress such as ischemia and nutrient deprivation [28,29]. In addition, it has also been proposed that dormant primordial follicles can be overactivated as a secondary effect. Chemotherapy damages developing follicles and thus removes their inhibitory effect and this results in the overactivation of primordial follicles in an attempt to replace the damaged cohort of growing follicles [25,26,77]. Each of these hypotheses will be introduced in the following sections.

### 3.1. Apoptosis Triggered by DNA Damage and/or Oxidative Stress

Cytostatic drugs are known to induce large amounts of DNA damage, and this is particularly true when alkylating agents are used [78]. Diplotene-arrested oocytes of primordial follicles are more sensitive to DNA damage compared to oocytes at other stages. There might be a stringent quality control surveillance system in primordial follicles [79]. Once DNA damage occurs, the oocytes are capable of repairing DNA DSBs, but there are fewer DNA repair responses in primordial follicles compared to developing follicles [79]. Unrepaired DNA damage triggers apoptosis in primordial follicles via TAp63. TAp63, which is a homologue of the p53 tumor suppressor protein, is specifically expressed in oocytes of primordial follicles, and its phosphorylation causes rapid apoptosis in response to DNA damage [21,80].

In vivo, cyclophosphamide (CPA) is activated by CYP2B and CYP3A to produce 4-hydroxycyclophosphamide (4-OH-CPA), which is subsequently interconverted and metabolized to phosphoramide mustard (PM) [81]. Experiments using cultured PD4 mouse ovarian tissue treated with PM showed a concentration-dependent increase of γH2AX, which is a marker of DSBs [23]. PM was shown to induce the activation of ataxia telangiectasia mutated (ATM), which is crucial in regulating and stimulating DSB repair, activating cell cycle checkpoints, and signaling for apoptosis induction [22,82]. The activation of ATM in these experiments indicates the occurrence of DNA damage after exposure to PM [22]. DNA damage-induced cell death has also been demonstrated in oocytes of cultured PD5 mouse ovaries by a concentration-dependent increase in γH2AX after prolonged exposure to different concentrations of cisplatin. In mouse oocytes, cisplatin treatment activated the c-Abl-TAp63 pathway by inducing the p63-dependent activation of pro-apoptotic promoters and by promoting TAp63 accumulation [21].

DNA damage and cellular stress such as oxidative stress interfere with the normal levels of pro-apoptotic and pro-survival molecules in cells [83]. Pro-apoptotic BH3-only proteins, especially PUMA and NOXA, are critical sensors of DNA damage and cellular stresses and trigger apoptosis [84,85]. It has been suggested that DNA damage-induced apoptosis in primordial follicle oocytes requires TAp63-mediated induction of PUMA and NOXA [85,86,87]. Primordial follicle depletion in mice exposed to cisplatin is reduced when TAp63 is lacking; however, this effect was not evident when CPA was studied under similar experimental conditions. Nevertheless, the crucial role of PUMA in primordial follicle depletion following exposure to CPA or cisplatin was demonstrated using *Puma^−/−^* mice [88].

For in vitro experiments, 4-hydroperoxycyclophosphamide (4-HC), a congener of CPA, can be converted into 4-OH-CPA without any enzymatic reactions, and 4-HC is much more readily transported into cells than PM [89]. A recent study proposed another apoptotic pathway for how 4-HC kills primordial oocytes in three independent murine strains at different ages. It was found that 4-HC killed primordial oocytes by inducing the p-ATR–p-CHK1/p-CHK2–p-p63–cPARP pathway without interfering with the surrounding pregranulosa cells or stromal cells, while at the same dose all granulosa cells of secondary follicles were killed. That study also indicated that various doses of CPA administration directly kill granulosa cells of growing follicles, including multilayer follicles, but do not lead to the death of these growing follicles [89].

Oxidative stress results from the imbalance between the systemic production of reactive oxygen species (ROS) and the ability to detoxify them or to repair the resulting damage. ROS disrupt normal cellular signaling by acting as cellular messengers, and oxidative stress causes DNA damage such as base modifications and strand breaks. Oxidative stress has thus been shown to have a significant negative impact on ovarian cells and oocyte health [90]. Some chemotherapeutic drugs generate oxidative metabolites or consume the antioxidant enzymes in the body, which increases oxidative stress and consequently triggers apoptosis in the ovary.

Active metabolites of CPA, such as PM and acrolein, are detoxified in the cells by conjugation with glutathione (GSH). Thus, the balance of GSH is important in the induction/reduction of oxidative stress. Treatment of COV434 granulosa cells with 4-HC caused the rapid depletion of intracellular GSH and increased ROS, which was followed by induction of apoptosis, and this effect was significantly suppressed by GSH supplementation or by co-treatment with antioxidants [24]. Exposure of mouse oocytes to acrolein in vitro also induced oxidative stress by generating ROS that further deteriorated the oocyte quality through alterations in the microtubule spindle structure and subsequent alterations in chromosome alignment [91].

### 3.2. Activation of Primordial Follicles and Ovarian Reserve Burnout

It has been hypothesized that chemotherapy-induced primordial follicle depletion might be due to the overactivation of primordial follicles from the dormant follicle pool. When the ovary is exposed to chemotherapeutic agents, the secretion of primordial follicle inhibitors decreases due to damage to the developing follicles, and this accelerates the recruitment of primordial follicles and the reduction of the ovarian reserve resulting in a “burnout” effect [25,26].

This has been supported by several studies showing decreased numbers of primordial follicles and increased numbers of early growing follicles in chemotherapy-treated groups compared with untreated groups in both mice and in cultured human ovarian tissue [25,26,92]. Analysis of key proteins showed that cisplatin treatment in mice activated the PI3K/PTEN/Akt/FOXO3a signaling pathway [25]. This theory was further supported by another study in which AMH expression levels dropped to below control level at 12–72 h after CPA treatment due to the loss of growing follicles. Additionally, the increase in early growing follicles was supported by a 2-fold increase in relative AMH expression 3–7 days after the treatment, and this increased relative expression was maintained until at least 14 days post-treatment [93]. It has also been demonstrated that blocking the primordial follicle recruitment pathway prevents the overactivation of follicles triggered by chemotherapy [26,94].

However, a study using three independent mouse strains at different ages did not find any significant increase in the number of primary or secondary follicles after various doses of CPA treatment. The authors concluded that the significant increase in the ratio of growing follicles to dormant follicles was due to a dramatic loss of dormant follicles while the number of growing follicles remained the same [89]. In that study, a decrease in AMH serum levels was observed within 3 days after CPA administration, but this returned to similar levels as in vehicle-treated mice by 7 days post-treatment. A possible explanation for this is that the decrease in AMH serum level is due to damage to the granulosa cells of growing follicles and that the ensuing increase is not due to the activation of dormant primordial follicles but to the replacement of damaged granulosa cells by healthy granulosa cells [89].

### 3.3. Inducing Autophagy

Oxidative stress, nutrient deprivation, hypoxia, and damaged organelles all induce apoptosis in the ovary and are also factors that trigger autophagy. Similar to apoptosis, autophagy appears to play an active role in follicle depletion in the ovary [95]. Cultured rat ovaries exposed to PM showed abnormally large Golgi apparatuses and electron-dense mitochondria by transmission election microscopy. Increases in the autophagy-related protein BECN1 and in lysosome-associated membrane protein have also been detected after PM exposure [27]. However, more research is needed to clearly identify all steps involved in the autophagy pathway.

### 3.4. Micro-Vessel Network Damage

A commonly observed negative effect of chemotherapy on the ovary is damage to the micro-vessel network and subsequent decrease in blood supply, which can induce stress factors such as ischemia, nutrient deprivation, and hypoxia followed by DNA damage and apoptosis. In ovaries of young female cancer patients who were previously exposed to chemotherapy, reductions in ovarian blood flow and ovarian size [96], cortical blood vessel damage as well as loss of small vessel proliferation have been reported [28]. Both in in vitro cultured and xenografted human ovarian tissues, doxorubicin treatment has been shown to reduce vascular density compared to vehicle-treated controls. [97].

In summary, chemotherapy-induced ovarian damage involves multiple mechanisms, but it is not clear which of these is the most important or what signaling pathways play a role in this process. Further research is therefore needed to identify which molecules can be targeted to block the negative response to chemotherapeutic agents and thus preserve as many ovarian primordial follicles as possible during treatment.

## 4. Potential Protective Treatments to Reduce/Recover Chemotherapy-Induced Ovarian Damage

The first pharmaceutical treatment aiming at protecting the ovary from chemotherapy-induced damage was proposed by Romona and Simon in 1981 with the use of oral contraceptives [98]. Although the authors reported a protective effect, the sample consisted of only six patients and no control group. Since then, an increasing number of studies testing different types of substances to protect the ovaries from chemotherapy-induced damage have been reported. A summary of these is presented in Table 2.

### 4.1. Proposed Treatments to Protect Against Apoptosis in Ovarian Follicles

Because apoptosis has been shown to play a major role in chemotherapy-induced follicular depletion, substances working against the factors triggering apoptosis or substances blocking apoptotic pathways have been proposed as potential candidates to reduce or prevent chemotherapy-induced follicle loss. These substances and the proposed protective mechanisms are reviewed in this section.

#### 4.1.1. Antioxidants

Antioxidants have the potential to eliminate ROS and prevent the damage that they cause. Experiments using rats exposed to cisplatin and co-treatment with antioxidants such as bilberry, mesna, sildenafil citrate, and hydrogen-rich saline have shown protective effects in the ovary. Co-treatment with bilberry increased the activity of the antioxidant enzymes superoxide dismutase, catalase, glutathione-s-transferase (GST), and glutathione peroxidase and reduced malondialdehyde levels [99]. Similar results were obtained in a study using mesna. Additionally, in HePG2 cell line culture, mesna did not interfere the anti-tumor effect of cisplatin [101]. In sildenafil citrate co-treated rats, more primordial follicles and higher immunoreactivity intensity of AMH were reported [102]. In rats, cisplatin-induced oxidative stress was also reversed by long-term injection of hydrogen-rich saline, thus showing lower FSH release, elevated estradiol, improved follicle development, and reduced ovarian cortex damage when compared to the group only treated with cisplatin. In addition, hydrogen-rich saline regulated the expression of a key regulator of the cellular antioxidant response – nuclear factor erythroid 2–related factor 2 (Nrf2)– in rats with ovarian damage [103,118].

Mirtazapine and hesperidin-treated rats showed significantly lower nitric oxide and malondialdehyde levels, lower myeloperoxidase activity, and significantly higher superoxide dismutase and glutathione peroxidase activity when compared with CPA-treated rats [100].

#### 4.1.2. Improving DNA Repair

Hypothetically, if some substances can prevent DNA damage or enhance DNA repair in follicles exposed to chemotherapeutic drugs before apoptosis is triggered, those follicles might be preserved.

Dexrazoxane is widely used in the clinic to reduce doxorubicin-induced heart and skin toxicity and has been shown to ameliorate doxorubicin-induced ovary injury by reducing DSBs and improving primary granulosa cell viability in cultured mouse and marmoset ovaries as well as in mouse granulosa cell lines [104,105]. Dexrazoxane provided not only acute protection, but also long-term protection [106]. Additionally, dexrazoxane did not interfere with the anti-tumor effects of doxorubicin nor did it increase the risk of secondary malignant neoplasm in leukemia cases [105,119,120,121,122,123,124,125,126,127].

#### 4.1.3. Mediating the Nuclear Accumulation of Chemotherapeutic Drugs

Bortezomib, a proteasome inhibitor, has been shown to reduce doxorubicin- induced ovary damage by mediating the nuclear accumulation of doxorubicin. Bortezomib competes with doxorubicin for binding to the proteasome and thus prevents the nuclear accumulation of doxorubicin by reducing the subsequent DNA damage and apoptosis. In mice co-treated with bortezomib, reductions in DNA damage, γH2AFX phosphorylation, and apoptosis were observed in preantral follicles when compared to doxorubicin group. Additionally, mice mated four weeks post-injection showed a reduced litter size in both groups; however, the group that received co-treatment with bortezomib recovered their litter size over time and improved the weight of the pups compared to the group that received only doxorubicin [107].

#### 4.1.4. Blocking the Apoptosis Pathway

Many molecules are involved in the apoptosis pathway, and up-regulation or down-regulation of these factors determines the fate of the cell. Within the follicle, chemotherapy can induce oocyte and/or granulosa apoptosis. Thus, substances inhibiting molecules involved in the apoptosis pathway, either in the oocyte or in granulosa cells, could have a protective effect on the follicles. However, it should be taken into consideration that different drugs induce apoptosis through different pathways. Thus, the inhibitory substance should be selected accordingly. Here, we present some substances that have been shown to have a protective effect in the ovary against the apoptotic mechanisms induced by chemotherapy drugs.

Imatinib, a c-Abl kinase inhibitor, has been shown to prevent the accumulation of c-Abl/TAp63 in mouse oocytes and thus to protect them from cisplatin-induced apoptosis [21]. However, this effect was only observed in cisplatin-treated mice because other chemotherapeutic drugs such as doxorubicin and platinum induce apoptosis in the ovary through other mechanisms [108,109].

KU55933, an inhibitor of the serine/threonine protein kinase ATM, has been shown to protect follicles of all stages from PM-induced depletion in PD4 rat ovaries by inhibiting ATM activation as part of the DNA damage response [22]. ATM is recruited and activated by DSBs and is suggested to play a key role in determining survival and death following genotoxic exposure through regulating or stimulating DSB repair, activating cell cycle checkpoints and inducing apoptosis [128].

Sphingosine-1-phosphate (S1P) is a metabolite of ceramide and an inhibitor of the ceramide-promoted apoptotic pathway [129]. In mice receiving an ovarian injection of S1P prior to busulfan treatment, significantly lower caspase-3 immunoreactivity, more primordial follicles, and higher AMH mRNA levels were observed compared to those only treated with busulfan. Thus, S1P has shown a protective effect against the gonadotoxicity of busulfan in the ovaries [110]. Significantly decreased levels of caspase-3 and reduced percentages of apoptotic follicles were also observed in xenografts of human ovarian tissue fragments receiving continuous infusion of S1P by a mini-osmotic pump 24 h before CPA or doxorubicin single dose treatment until 72 h after chemotherapy [111].

ETP46464 is a highly specific inhibitor of ATR, and CK2II is a CHK2 inhibitor. In cultured PD5 mouse ovaries, ETP46464 or CK2II pretreatment effectively protected primordial follicles from 4-HC-induced loss and decreased the presence of p-p63α in primordial follicles. However, neither of these inhibitors could protect granulosa cells of multilayer secondary follicles from 4-HC-induced death [89].

### 4.2. Proposed Treatments to Reduce Overactivation of the Primordial Follicle Pool

AMH, which is produced by the granulosa cells of growing follicles, has been shown to inhibit the initiation of primordial follicle growth [46,60]. Therefore, the death of growing follicles after chemotherapy results in a decrease in the levels of AMH in the ovary and a subsequent overactivation of primordial follicles has been postulated.

In mice treated with supra-physiological doses of AMH, the complete arrest of folliculogenesis with a contraceptive effect was obtained after long-term parenteral AMH administration. This effect was reversed after the treatment was stopped, and significantly more primordial follicles were detected in mice co-treated with AMH concomitantly with chemotherapy, compared to those treated only by chemotherapy [112]. In addition, AMH is a natural endogenous hormone with a unique mechanism of action in the ovary that makes it particularly suited for use during chemotherapy.

The immunomodulator ammonium-trichloro (0,0-dioxyethylene) tellurate (AS101) has shown similar effects by reducing primordial follicle activation after CPA treatment in mouse ovaries. In addition, it has been shown in breast cancer cell lines that rather than decreasing the anti-tumor effect of CPA, AS101 increased its anti-tumor efficacy [26].

Activation of the PI3K signaling pathway activates the mTOR signaling pathway through elevated Akt levels. Chemotherapy such as CPA can cause the activation of these pathways, and administration of the mTORC1 inhibitor rapamycin has been shown to prevent the overactivation of the primordial follicles in the follicle pool and to preserve fertility [27,94].

Melatonin has been shown to significantly decrease cisplatin-induced overactivation of the dormant primordial follicle reserve by increasing phosphorylation of PTEN and consequently inhibiting the activation of Akt, GSK3β, and FOXO3a in the mouse ovary [113].

Gonadotropin-releasing hormone (GnRH) is a decapeptide produced by the hypothalamus and is responsible for the release of FSH and LH by the pituitary gland, which regulate ovarian follicle development and maturation. Continuous GnRH production, which can be achieved using GnRH analogues (GnRHa), induces a reduction in receptor sensitivity, resulting in a progressive decrease in FSH and LH levels. Thus, ovarian function is suppressed and it enters a quiescent state or a state similar that of the pre-pubescent ovary. It has been proposed that the ovary in this state might be less sensitive to chemotherapy-induced damage [130]. The effect of GnRHa in the ovary aiming at reducing chemotherapy-induced damage has been proposed as ovarian protective during chemotherapy and it has been investigated over almost three decades in both animal experiments and human clinical trials, with still inconclusive results.

In some animal experiments, mice treated with triptorelin, a GnRHa, showed dose-dependent protection against busulfan or CPA-induced follicle depletion [114,131]. Similarly in mice, cetrorelix, a GnRH antagonist, significantly decreased the extent of ovarian damage induced by CPA [132]. However, there is still some controversy regarding these results, and there are several animal studies suggesting that GnRHa does not provide a protective effect in the ovary during chemotherapy [133,134,135]. In a recent study, *Fshb*^−/−^ mice were used to mimic the profound inhibition of FSH secretion during GnRH analogues treatment. In *Fshb*^−/−^ mice, CPA induced significant follicle loss but did not cause any change in cell proliferation or apoptosis regardless of prior treatment with GnRHa or saline. Furthermore, in cultured CBA/ca × C57Bl mouse preantral ovarian follicles, 4-HC significantly delayed follicular development and decreased the survival and maturation rate both in GnRHa-treated and untreated control groups. Similarly, exposure of cultured neonatal CBA/ca × C57Bl mouse ovaries to 4-HC induced apoptosis and significant follicular loss in both the GnRHa-treated and untreated control groups [136]. Another in vitro study using human ovarian cortical pieces treated with leuprolide acetate concomitantly with different chemotherapeutic agents including CPA, cisplatin, paclitaxel, fluorouracil, or TAC combination regimen (docetaxel, adriamycin, and CPA) did not show activation of any anti-apoptotic pathways in the leuprolide groups, and the follicle loss, DNA damage, and apoptosis induced by these drugs were not prevented [137]. In vivo co-administration of CPA and GnRHa to rats also showed no protective effect on DNA damage compared to CPA treated group [138].

In humans, GnRHa has been investigated in numerous cohort studies as well as in a limited number of clinical trials, as summarized in a recent meta-analysis [139]. Although the results of these analyses indicated greater resumption of menses in pre-menopausal patients, fertility preservation was a pre-planned secondary endpoint in only one of the five included trials [140]. Because none of the reported trials were blinded or placebo controlled, the GnRHa-treated women might have been more likely to attempt pregnancy, as illustrated by the only trial that recorded attempts to pregnancy in the study participants [140]. A previous meta-analysis of six studies that included clinically established markers of ovarian reserve, as estimated by the hormone dosage of AMH, which is the gold standard for this estimation, reported no improvement in ovarian markers though higher rates of recovery of menses in the GnRHa group, and no differences in pregnancy rates or fertility were found between the groups [141]. In line with these results, a recent Cochrane systematic review pointed out that there was insufficient evidence of fertility protection from GnRHa [142]. Of note, the 2018 panel of the American Society of Clinical Oncology, ASCO, recommended that GnRHa should not be used in place of proven fertility preservation methods [143].

Tamoxifen, a selective estrogen receptor modulator that might increase gonadotropin levels (mainly FSH secretion), has also been proposed to reduce CPA-induced follicle damage. In animal experiments, tamoxifen has been shown to block CPA-induced follicular toxicity and to improve the survival rate of offspring in rats after CPA treatment, as well as to prevent oocyte fragmentation from doxorubicin exposure in vitro [144].

### 4.3. Decreased Micro-Vessel Loss in the Ovary

Mice co-treated with CPA, busulfan, and granulocyte colony-stimulating factor (G-CSF) had a 10-fold greater follicular reserve, increased micro-vessel density, and extended reproductive lifespan compared to mice treated only with CPA and busulfan. In addition, the co-treated groups had similar levels of phospho-γH2AX-induced damage as untreated groups. This protective effect of G-CSF was mainly ascribed to decreased blood vessel loss and ischemia following chemotherapy, which may result in less vascular injury and/or stimulation of new blood vessel formation that better supports the survival of ovarian follicles [117]. Additionally, G-CSF was successfully used in cancer patients to prevent neutropenia caused by chemotherapy without decreasing chemotherapy efficacy [145]. Therefore, G-CSF might be a safe ovary protectant that does not interfere with the anti-tumor effect of chemotherapy, although more studies are needed to better illustrate its protective effect in the ovary during chemotherapy.

### 4.4. Others

The potential ovarian-protection methods introduced in this section are not specifically related to working against chemotherapy-induced damage to the ovary, but are instead generally seeking to change the chemotherapy recipient or to introduce modifications to the chemotherapeutic drug itself.

#### 4.4.1. Caloric Restriction

Obesity seems to compromise the DNA repair response by altering DNA repair response-related mRNA and protein expression [146]. Researchers found that progressive obesity in mice could alter the mRNA expression of genes involved in ovarian insulin-mediated PI3K signaling, e.g. *Akt* and *Foxo3a*, as well as influence the mRNA expression of genes involved in xenobiotic biotransformation, e.g. *Gst* and *Cyp450*, which potentiate the ovotoxicity induced by PM [147].

Rats undergoing a 65% calorie restriction for eight weeks showed better ovarian follicular reserve after CPA treatment compared to ad libitum-fed controls, especially in terms of primordial and primary follicles [148]. This might be related to an increase in the longevity protein SIRT1 and a decrease in p53, and *SIRT1*-deficient cells are more susceptible to apoptosis induced by cisplatin [149].

#### 4.4.2. Gene Therapy

Transduction of the multidrug resistance gene (*MDR1*) into follicles can help them avoid chemotherapy-induced damage. The KK15 cell line is an immortalized murine granulosa cell line that after transduction can express high levels of biologically active *MDR1* and has shown greater tolerance to doxorubicin and paclitaxel treatment and higher cell viability [150]. Nevertheless, *MDR1* is also a way by which tumor cells escape from chemotherapy, so the safety of using *MDR1* transduction in follicles during cancer treatment must be ensured. In addition, follicles are not cells, and to achieve the multidrug resistance of follicles the gene should be transduced into both the oocyte and the granulosa cells.

#### 4.4.3. Pharmaceutical Development and Improvement of Chemotherapeutic Treatment

Modifications to chemotherapeutic treatments such as targeted delivery systems can not only improve the delivery of chemotherapeutic drugs (reducing drug lose during delivery), but also improve the specificity of tumor-targeted treatments, thus avoiding the off-targeted effects that current treatments bring to most organs and systems. In the future, these treatment modifications will benefit cancer patients by reducing gonadal toxicity, thus prevent endocrine deficiencies and POI.

Improving arsenic trioxide delivery by using a nanoscale formulation has been shown to be much less deleterious to ovarian function than the parent drug in an in vitro assay of ovarian follicle function that can predict the in vivo ovarian toxicity of therapeutic agents [151].

### 4.5. Repair or Replace Damaged Cells in the Ovary After Chemotherapy

The methods introduced in this section do not provide ovarian protection during chemotherapy, but instead focus on repairing or replacing damaged cells after chemotherapy.

#### 4.5.1. Human Amniotic Fluid Cells (hAFCs) and Human Amniotic Epithelial Cells (hAECs)

Transplantation of hAFCs cloned and cultured in vitro into the ovaries of mice sterilized by intraperitoneal injection of CPA and busulfan restored ovarian function. By tracing and detecting the marked grafted hAFCs, it was shown that they survived and differentiated to granulosa cells. Compared to sterilized control mice, there were observable follicle-enclosed oocytes at all stages of development and stronger AMH expressions in hAFCs treated ovaries [152]. Similarly, hAECs could be cultured in vitro, and after being intravenously injected into CPA and busulfan-sterilized mice they migrated to the ovary, differentiated into granulosa cells, and restored follicle development [153].

Despite these successes, in these cases there will be questions of who can be donors, and more studies are needed to clearly illustrate the mechanisms through which these cells exert their effects in order to make them more feasible for clinical use.

#### 4.5.2. Bone Marrow-Derived Mesenchymal Stem Cells (BMMSCs)

Female rats received intraperitoneal injections of CPA followed by male BMMSCs intraperitoneal injections showed increased primordial follicle count and decreased caspase-9 and TUNEL levels compared with those only treated with CPA [138]. The detection of the *Sry* gene in the ovarian tissues of the co-treated group demonstrated that BMMSCs had migrated to the injured ovaries, accelerated the repair response, and decreased DNA damage-induced apoptosis. However, because stem cells are pluripotent, more studies are needed to clearly illustrate how these BMMSCs behave in the recipient in order to exclude the possibility of them differentiating into oocytes, in which case the offspring might biologically belong to the donor.

### 4.6. Other Possible Protective Substance Resources

Due to the unselective toxic effects of chemotherapy to healthy cells and tissues in organs such as in the heart, the gastrointestinal tract, liver and kidney, current research on organ-protective reagents specific to those organs could provide good potential candidates for pharmacological ovarian protection.

For example, a small PUMA inhibitor molecule has shown protective effects against chemotherapy-induced damage both in mouse and human colonic organoids, without protecting cancer cells [154]. Additionally, in a recent study, the RRx-001, a next generation checkpoint inhibitor with minimal toxicity, showed selective protection of normal cells against cisplatin-induced cell death through Nrf2 induction; however apoptosis of tumor cells was induced by inhibiting Bcl-2 [155].

## 5. Conclusions and Perspectives

This review summarizes current knowledge on chemotherapy-induced ovarian damage, the proposed mechanisms involved in the damage, and several potential protective methods aiming at protecting ovarian health and germ line integrity that have been demonstrated to partially or completely circumvent the damage. During the writing of the review we found the difficulties, complexities and uncertainties lying on the way of achieved protectants to go from laboratory until their clinical use. 

A problematic issue is the difficulty in evaluating the clinical efficacy of the protection. Since the size of the ovarian follicle reserve after chemotherapy determines the ability of being fertile in the future, for some women it could be clinically assessed, using markers of ovarian reserve, that their fertility potential is reduced. However, it is difficult to say what is the minimum size of the follicle pool to fulfill fertility. Maybe in some women, a small proportion of primordial follicles can eventually give some mature follicles, but in other women they might not. Most of the studies summarized in this review have shown an attenuation in chemotherapy-induced ovarian damage; however, the studies show that the ovarian reserve was still decreased comparing to the non-treated groups. In other words, the potential protectants had the ability to reduce the damage, rather than to prevent it. Whether this attenuation is strong enough to ensure the ability of being fertile in the future is uncertain.

A problem of outmost importance to be solved is the safety of using cell-protectants, especially in the context of cancer patients treated for malignant diseases. Since chemotherapy is genotoxic, the oocytes of primordial follicles that we eventually want to protect will bear the responsibility of forming half of the genes of the next generation. Even a single change in the genes of the oocyte can cause a tremendous tragedy to the next generation or even further generations by cumulative inheritance. We have to say that the disappearance of primordial follicles after chemotherapy is a well-designed self-monitoring mechanism of the body to avoid the damaged genes from being passed down to the next generation. If the protectants force the primordial follicles with damaged genes to survive, the health of the oocytes cannot be ensured. Furthermore, in clinical cancer treatment, the combination of different types of chemotherapy is very common; they work through different mechanisms and thus may need corresponding specific protectants, which results in the need to develop a combination of protectants specifically adapted to the patient’s chemotherapy.

Additional difficulties include the delay in translation. Even though breeding experiments using animal models after chemotherapy have proved successful pregnancies, resulting in healthy pups, the species difference and the inter-individual differences in humans are all problems that need to be considered. Additionally, as reviewed, chemotherapy-induced damage involves a plethora of molecules and pathways, but the up- and down-stream relationship and the importance among them is mostly unknown, making it difficult to find the key point. That can also explain why several substances have been shown to have partial protective efficacy instead of total prevention. Last but not least, it is difficult to investigate the efficacy of one protectant, since the timing and the dose of the protectant may determine the final outcome. A correct dose of the substance at the wrong time point in association with the chemotherapeutic treatment might result in no protection; however, it may be too early to draw the conclusion that the substance is not protective.

Even though there are those difficulties, complexities and uncertainties, we still expect the translation of fertility protective treatment to come to the clinic in the near future. With rigorous evaluations and confirmations, well-designed clinical trials and individualized formulae, we believe the questions of efficacy and safety can be well addressed and fertility protective treatment can be approved as a therapeutic intervention during chemotherapy.

## Figures and Tables

**Figure 1 ijms-20-04720-f001:**
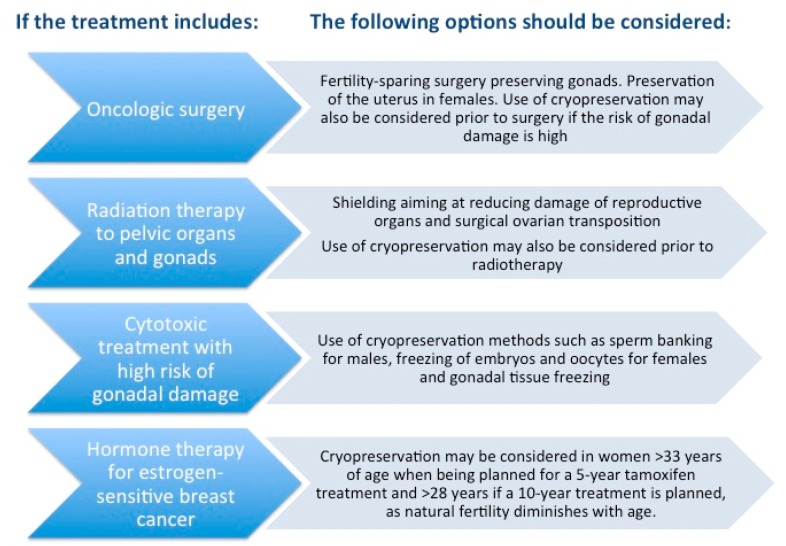
Fertility preservation strategies depending on the type of oncological treatment in females and males. Reprinted, with permission from Rodriguez-Wallberg and Oktay. Fertility preservation during cancer treatment: Clinical guidelines. Cancer Management and Research, 2014:6 105–117 [4].

**Figure 2 ijms-20-04720-f002:**
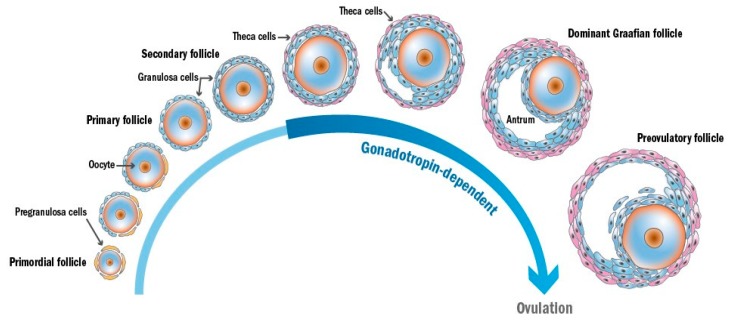
Different stages of follicular development.

**Figure 3 ijms-20-04720-f003:**
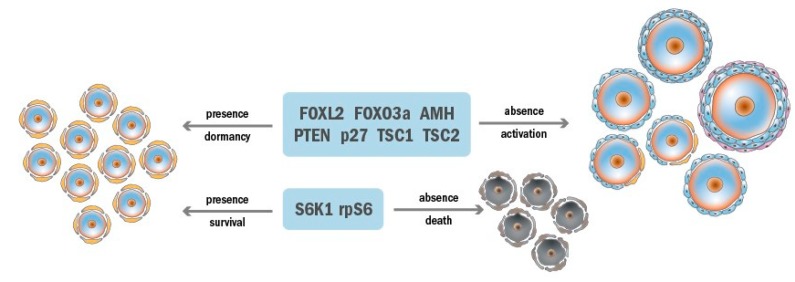
Factors maintaining the dormancy and survival of primordial follicles.

**Figure 4 ijms-20-04720-f004:**
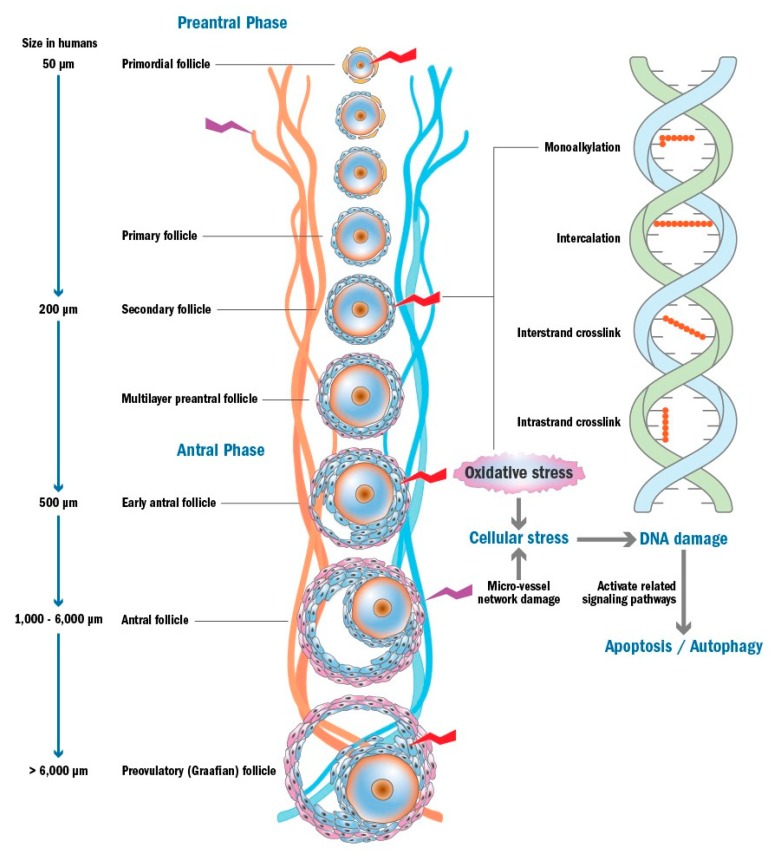
Proposed mechanisms of chemotherapy-induced ovarian damage and follicle depletion.

**Table 1 ijms-20-04720-t001:** Types of chemotherapeutic agents and their corresponding antitumor mechanisms.

Type of Agents	Representatives	Anti-Tumor Mechanisms	Targeting Cell Cycle
Alkylating agents	Cyclophosphamide, iphosphamide, melphalan, busulfan, nitrogen mustard, nitrosoureas, procarbazine, chlorambucil	Produce highly reactive intermediates that form covalent bonds with nucleophilic substances; cause DNA inter- and intra-chain cross links; interfere with DNA transcription and replication	Non-specific
Platinum analogs	Cisplatin, carboplatin	Crosslink with purine bases; cause DNA damage; and interfere with DNA repair	Non-specific
Taxanes and Plant alkaloids	Paclitaxel, docetaxel, vincristine	Bind tubulin to inhibit its polymerization into microtubules; prevent spindle formation; and cause metaphase arrest	M-phase specific
Antitumor Antibiotics	Mitomycin, bleomycin, doxorubicin, valrubicin	Intercalate into the minor groove of double-stranded DNA between guanine-cytosine base pairs; interfere with RNA polymerase movement along the DNA; prevent transcription; cause DNA double-strand breaks; and stabilize DNA-topoisomerase II complexes	Specific and non-specific
Topoisomerase inhibitors	Epipodophyllotoxins etoposide, teniposide	Inhibit topoisomerase; suppress microtubule aggregation; and inhibit spindle formation	S- and G2-phase specific
Antimetabolites	Methotrexate, 5-fluorouracil, 6-mercaptopurine, hydroxyurea,	Inhibit the synthesis of or compete with purine or pyrimidine nucleotide precursors during DNA or RNA synthesis	S-phase specific
Enzymes	l-asparaginase	Inhibit the enzyme ribonucleotide diphosphate reductase; limit ribonucleotide conversion thus block DNA synthesis; deprive exogenously supplied asparagine thus limits protein synthesis	S-phase specific

**Table 2 ijms-20-04720-t002:** Substances used to prevent chemotherapy-induced ovarian damage.

Group	Mechanism Proposed	Substance	Chemotherapy	Experimental Model	Reference
Anti-oxidants	Alleviate free radical damage	Bilberry	Cisplatin	In vivo; rat	[99]
		Mirtazapine or Hesperidin	CPA	In vivo; rat	[100]
		Mesna	Cisplatin	In vivo; rat	[101]
		Sildenafil citrate	Cisplatin	In vivo; rat	[102]
		Hydrogen-rich saline	Cisplatin	In vivo; rat	[103]
Iron chelate drugs	Reduce the number of metal ions complexed with anthracycline decreasing the formation of superoxide radicals	Dexrazoxane	Doxorubicin	In vitro; marmoset ovarian tissue	[104]
				In vitro; mouse cell line and mouse ovary	[105]
				In vivo; mouse	[106]
Proteasome inhibitors	Inhibit chemotherapeutic drugs’ nuclear accumulation	Bortezomib	Doxorubicin	In vivo; mouse	[107]
c -Abl kinase inhibitors	Block apoptosis pathway	Imatinib	Cisplatin	In vitro, mouse ovary; in vivo, mouse ovary sub-renal graft	[21,108,109]
ATM inhibitors		KU55933	CPA	In vitro; rat ovary	[22]
Ceramide-induced death pathway inhibitors		S1P	Busulfan,	In vivo; mouse	[110]
			CPA, doxorubicin	In vivo; human ovarian xenograft to mouse	[111]
CHK2 inhibitors		CK2II	4-HC	In vitro; mouse	[89]
ATR inhibitors		ETP46464	4-HC	In vitro; mouse	[89]
Glycoprotein hormones	Inhibit primordial follicle overactivation	AMH	Carboplatin, doxorubicin, CPA	In vivo; mouse	[112]
Immunomodulators		AS101	CPA	In vivo; mouse	[26]
mTOR inhibitors		Rapamycin	CPA	In vitro; mouse ovary	[27]
Hormones & Free radical scavengers		Melatonin	Cisplatin	In vivo; mouse	[113]
GnRHa		Triptorelin	Busulfan	In vivo; mouse	[114]
		Leuprolide acetate	CPA, doxorubicin	Breast cancer patients	[115]
		Goserelin	CPA, anthracycline	Breast cancer patients	[116]
Colony and stem cell factors	Decrease ovarian micro-vessel loss	G-CSF ± SCF	CPA, busulfan	In vivo; mouse	[117]

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
