# Peer review of "Ovarian Follicle Depletion Induced by Chemotherapy and the Investigational Stages of Potential Fertility-Protective Treatments—A Review"

_ijms, 2019, doi:10.3390/ijms20194720_

Round 1

Reviewer 1 Report

I think this review has an excellent introduction and a very detailed list of potentially protecting agents,

what I miss is in the "conclusions and perspectives"  a discussion prepared with the same authoritative capacity that the Authors display in the introduction to give their opinion on which compounds are closer to clinical use and/or more likely to provide a relevant protection.

There are some minor mistakes in table 1:

mitomycin and bleomycin are not  anthracyclines

Hydroxyurea is not an enzyme

Author Response

Comments and Suggestions for Authors

I think this review has an excellent introduction and a very detailed list of potentially protecting agents,

what I miss is in the "conclusions and perspectives" a discussion prepared with the same authoritative capacity that the Authors display in the introduction to give their opinion on which compounds are closer to clinical use and/or more likely to provide a relevant protection.

There are some minor mistakes in table 1:

mitomycin and bleomycin are not anthracyclines

Hydroxyurea is not an enzyme

Response: Thank you for your constructive comments and suggestions. We have added our opinion to the "conclusions and perspectives" section, including a comment about the future clinical use of the protectants. We have also corrected the typos in table 1, as indicated. They were caused by an unplanned shift in the table columns, thus the description of the drugs were not aligned with the drugs' names.

Reviewer 2 Report

Hao et al summarize in this review the current status of knowledge about the potential mechanisms of damage of chemotherapy on ovarian reserve. The article gives also an interesting overview of the possible protective treatments to reduce chemotherapy-induced ovarian damage. In this last decade, more attention has been given to fertility preservation since the improvements of cancer treatments have increased the survival rates and therefore more women face the sequelae of exposure to cytotoxic drug like premature ovarian insufficiency (POI) and infertility. Preventing follicle depletion would avoid not only POI related infertility but also the endocrine related side effects induced by premature ovarian failure (POF).  

This paper analyzes the proposed mechanisms behind the DNA-damage and overactivation of primordial follicles induced by chemotherapeutic drugs. The fourth paragraph systematically reports the published studies about different types of substances and experimental methods proposed to protect against ovarian damage.

At page 11, lines 387-389: it should be cleared better.

At page 13, line 473-484: a recent update of the Cochrane Systematic Review by Chen et al  (2019) still confirms that the evidence for protection of fertility by GnRH analogues is insufficient, while it seems effective in terms of maintenance and resumption of menstruation and treatment related POI.  In line with these results, 2018 ASCO Reccomendations still suggest that "GnRHa should not be used in place of proven fertility preservation methods".  

Paragraph 5 "Conclusions and Perspectives": this paragraph is difficult to read, especially lines 574-579. The concepts the authors want to express about the difficulties in predicting the level of ovarian damage and the fundamental aim of reducing rather than preventing it, are very important and deem a more exhaustive explanation and discussion. 

Overall, the main strength of this paper is the completeness of the information provided. Sometimes language is difficult to follow, some concepts should be better explained and a quick copy edit is suggested.

Author Response

Comments and Suggestions for Authors

Hao et al summarize in this review the current status of knowledge about the potential mechanisms of damage of chemotherapy on ovarian reserve. The article gives also an interesting overview of the possible protective treatments to reduce chemotherapy-induced ovarian damage. In this last decade, more attention has been given to fertility preservation since the improvements of cancer treatments have increased the survival rates and therefore more women face the sequelae of exposure to cytotoxic drug like premature ovarian insufficiency (POI) and infertility. Preventing follicle depletion would avoid not only POI related infertility but also the endocrine related side effects induced by premature ovarian failure (POF).  

This paper analyzes the proposed mechanisms behind the DNA-damage and overactivation of primordial follicles induced by chemotherapeutic drugs. The fourth paragraph systematically reports the published studies about different types of substances and experimental methods proposed to protect against ovarian damage.

Specific comments:

At page 11, lines 387-389: it should be cleared better.

Response: Thank you. We now clarified the sentence, this text is currently on Lines 394-397.

At page 13, line 473-484: a recent update of the Cochrane Systematic Review by Chen et al  (2019) still confirms that the evidence for protection of fertility by GnRH analogues is insufficient, while it seems effective in terms of maintenance and resumption of menstruation and treatment related POI.  In line with these results, 2018 ASCO Reccomendations still suggest that "GnRHa should not be used in place of proven fertility preservation methods".  

Response: Thank you for this suggestion. We have now clarified the text and included these two references.

Paragraph 5 "Conclusions and Perspectives": this paragraph is difficult to read, especially lines 574-579. The concepts the authors want to express about the difficulties in predicting the level of ovarian damage and the fundamental aim of reducing rather than preventing it, are very important and deem a more exhaustive explanation and discussion. 

Overall, the main strength of this paper is the completeness of the information provided. Sometimes language is difficult to follow, some concepts should be better explained and a quick copy edit is suggested.

Response: Thank you. We have improved our section: “Conclusions and Perspectives” and the manuscript has been revised by a English native professional.